# Association between hypertension and self-perception of health status: Findings from a decade population-based survey in Spanish adults

Jesús Martín-Fernández [1,2,3,4*], Tamara Alonso-Safont[5,6], Elena Polentinos-Castro[2,3,4,7], Gemma Rodríguez-Martínez[8], Mª Isabel González-Anglada[2,9], Amaia Bilbao-González[3,10,11,12], Isabel del-Cura-González[2,3,4,7,13]

1 Oeste Family and Community Care Teaching Unit, Primary Care Assistance Management, Madrid Health Service, Madrid, Spain, 2 Department of Medical Specialties and Public Health, Faculty of Health Sciences, Rey Juan Carlos University, Madrid, Spain, 3 Research Network on Chronicity, Primary Care and Health Promotion-RICAPPS (RICORS), ISCIII, Madrid, Spain, 4 Gregorio Marañón Health Research Institute (IISGM), Madrid, Spain, 5 Technical Directorate of Health Information Systems, Primary Care Management, Madrid Health Service, Madrid, Spain, 6 Doctoral Program in Health Sciences, Rey Juan Carlos University, Madrid, Spain, 7 Research Unit, Primary Care Assistance Management, Madrid Health Service, Madrid, Spain, 8 Infante Don Luis Health Center, Primary Care Assistance Management, Madrid Health Service, Madrid, Spain, 9 Internal Medicine Service, Alcorcón Foundation University Hospital, Madrid, Spain, 10 Osakidetza Basque Health Service, Basurto University Hospital, Research and Innovation Unit, Bilbao, Spain, 11 Biosistemak Health Services Research Institute, Barakaldo, Spain, 12 Department of Medicine, Faculty of Health Sciences, University of Deusto, Bilbao, Spain, 13 Ageing Research Center. Karolinska Institute, Stockholm, Sweden

* jmfernandez@salud.madrid.org

## Abstract

### Objective

This study, conducted in the community setting, aimed to assess and discuss how a diagnosis of arterial hypertension affects self-perceived health status, examining the association with potential explanatory factors and comparing its impact with that of other chronic conditions.

### Methods

Cross-sectional observational study using the 2011–2012 and 2017 Spanish National Health Surveys and the 2020 European Health Interview Survey for Spain as data sources. Health perception was categorised as very good, good, fair, bad, or very bad. The independent variables recorded demographic, social, clinical, and lifestyle information. The associations between variables were evaluated via a generalisation of an ordered logit model.

### Results

A total of 66,168 subjects were included (21,007 in 2011, 23,089 in 2017, and 22,072 in 2020), 21.6% of whom were diagnosed with hypertension, 51.3% were women,

**Data availability statement:** All relevant data are within the manuscript and its Supporting Information files.

**Funding:** The translation of the manuscript was financed by the Instituto de Salud Carlos III (ISCIII, Research Project 18700370, co-funded by the European Union). The funders had no role in study design, data collection and analysis, decision to publish, or preparation of the manuscript.

**Competing interests:** The authors have declared that no competing interests exist

and the average age was 48.24 (18.89) years. Around one in five people in the general population reported a "very good" health status. The probability of reporting a "very good" health condition decreased with a diagnosis of hypertension (6.2%; CI 95%: 3.1–9.3%) and hypertensive medication (4.5%; CI 95%: 1.8–7.3%). Such associations were independent of age, gender, social group, other chronic conditions or limitations, or various lifestyle habits. In contrast, no association was found with reporting a "bad" or "very bad" health status.

## Conclusion

Being diagnosed with hypertension and prescription of antihypertensive medication are associated with a lower probability of reporting a "very good" health status, irrespective of other comorbidities or complications related to the diagnosis.

## Introduction

Arterial hypertension (HTN) is one of the most prevalent conditions in adults, reaching 32% of women and 34% of men over 30 years of age [1]. This entails a significant disease burden, making HTN responsible for the loss of 143 million disability-adjusted life years globally in 2015, an increase of > 30% versus those in 1990 [2]. This growing prevalence of HTN is partly explained by the increase in obesity and unhealthy lifestyle habits [3,4] as well as the population ageing, as subjects who reach the average life expectancy without HTN are > 90% more likely to develop the disease throughout their remaining life [5].

Numerous studies have demonstrated the association of HTN with various conditions, especially cardiovascular diseases [2,6,7] that lower life expectancy [8–11]. HTN is generally asymptomatic in the absence of comorbidities. This can lead patients to erroneously believe that the disease is benign, which entails implications when adopting permanent lifestyle [12] or pharmacological [13–15] recommendations.

Various constructs have been designed to understand the repercussions that a health situation or lack thereof has on the patient's life. Health-related quality of life (HRQoL) is one such construct, which can be defined as the self-perceived health condition and how disease or treatments affect the well-being and functionality of patients [16]. The concept of HRQoL overlaps with the self-perception of health status. Therefore, HRQoL could be defined as the way in which health affects quality of life or used to state a preference associated with a certain health status [17].

Some research has demonstrated a negative impact of hypertension (HTN) on health perception [18–20]. Trevisol et al., in a systematic review of over 20 studies, found that individuals with HTN reported slightly worse health perception compared to normotensive individuals. However, the authors noted that the role of hypertension and the awareness of having the condition requires further investigation[19]. Following this, the same author conducted a new population-based study, which

concluded that worse health perception was more pronounced in patients undergoing treatment, particularly when blood pressure was controlled by medication [21].

Other studies have shown that HTN alone significantly worsens health perception, with factors such as gender, education level, blood pressure control, and physical activity further influencing this effect. Additionally, drug-related factors, including the number of medications, disease duration, and adverse drug reactions, have been found to exacerbate the impact on health perception [22]. However, an observational study conducted in a European population found that the stage of HTN and awareness of the condition did not significantly affect physical or mental health. This suggests that the asymptomatic nature of early-stage hypertension could explain why it has a minimal impact on health perception [23].

In Spain, several studies have explored the relationship between hypertension and health perception, revealing a complex and not fully understood connection. A 2006 study with 3,368 participants aged 60 or older found no significant association between hypertension, antihypertensive treatment, or blood pressure control and worse health perception when assessed using the SF-36 questionnaire [24]. An other analysis conducted in individuals over 65 with multimorbidity found that hypertension was not linked to worse quality of life [25]. Similarly, research involving over 11,000 individuals showed that hypertension alone had minimal impact on health perception, but when combined with overweight, obesity, or other chronic conditions like diabetes, the likelihood of poor health perception increased significantly [26]. These findings suggest that hypertension's impact on health perception is limited unless accompanied by other health issues.

The perception of health status changes throughout life [27]. Therefore, such perception should be affected by HTN, as it becomes more prevalent with age. However, a "recalibration" phenomenon has been shown in hypertensive patients, in which a lower functional status is assumed as "normal" with ageing [28]. Therefore, the perceived health in older adults with HTN appears to be the result of a realistic appraisal of age, chronic diseases, recent acute events, functionality, and stress [29].

Whether the self-perception of health worsens in hypertensive patients is still unclear, and so is whether this alteration is due to the diagnosis, chronic medication, or comorbidities. The aim of this study conducted in the community setting was to assess and discuss how a diagnosis of arterial hypertension affects self-perceived health status, examining the association with potential explanatory factors and comparing its impact with that of other chronic conditions.

## Methods

### Design

An observational study was designed. We analyzed data collected from the Spanish National Health Survey (SNHS) in 2011–12 (n = 21,007), 2017 (n = 23,089), and from the European Health Interview Survey (EHIS) for Spain in 2020 (n = 22,082), which are anonymised and publically available (https://www.ine.es/dyngs/INEbase/es/categoria.htm?c=Estadistica_P&cid=1254735971047).

Stratified sampling was performed in three phases as part of the survey methodology: first, census sections were selected (2,000 sections in 2011 and 2,500 sections in 2017 and 2020); second, 12 households were chosen from each section in 2011, and 15 households from each section in 2017 and 2020; third, one adult aged 15 years or older was selected from each household. The only selection criteria were being 15 years or older and living in the selected household. All individuals meeting this condition had an equal probability of being selected, but only one person was included from each household. The sample size for each survey was calculated to ensure representativeness at the regional level (Autonomous Community) [30–32].

### Variables

The outcome variables were collected through equally formulated questions from the three surveys. The dependent variable was self-perceived health status, which was categorised into five levels: very good, good, fair, bad, and very bad. The independent variables were the hypertension diagnosis together with demographic, social, clinical, and lifestyle data.

Age, gender, and residence region were recorded as demographic variables. The social data included nationality (Spanish/foreigner), marital status (single, married, widow/separated/divorced), and social class in six levels based on the professional category. We designated "high class" as encompassing levels I and II, which include directors and managers of business establishments, professionals with university-level education, and athletes and artists [33].

Lifestyle-related variables were also collected, such as smoking habit (current versus former smoker/never smoked), physical activity (regular if performed several times a month), and body mass index (BMI) (underweight, normal weight, overweight, and obese, comparing normal weight versus the rest).

The recorded clinical variables were the existence of a HTN diagnosis, prescription for HTN, other cardiovascular pathologies (myocardial infarction, heart pathologies, or cerebrovascular accidents), osteoarticular pathologies (arthrosis, arthritis, rheumatism, or chronic back pain), respiratory disease (asthma or chronic bronchitis, emphysema, chronic obstructive pulmonary disease), chronic mental disorder (depression, anxiety, or other conditions), and recent diagnosis of anxiety or depression (last 12 months).

The subject's perception of limitation due to illness (severe, mild, and no limitation, comparing the latter versus the first two) was also included. The number of prescribed chronic drugs other than antihypertensive medication was obtained and polypharmacy was considered present for $\geq 5$ different types of treatment.

## Analysis

The variable distributions were described by absolute values and frequencies in the case of qualitative ones and by measures of central tendency and dispersion in the case of continuous ones.

The self-reported perception of health status was analysed via a five-category variable: very good ($Y=1$), good ($Y=2$), fair ($Y=3$), bad ($Y=4$), and very bad ($Y=5$). As these are ordered according to the level of health status (inversely), a standard ordered Logit model was selected. The probability that Y is included in a category is calculated as [34]:

$$P(Y=1) = 1 - g(X\beta_1)$$

$$P(Y=m) = g(X\beta_{m-1}) - g(X\beta_m), m = 2, \ldots, J-1$$

$$P(Y=J) = g(X\beta_{J-1})$$

where X represents the matrix of covariates and β the vector of coefficients. The term Y represents the discrete categorisation of a non-observable continuous latent variable Y*, which stands for the real perception of health status and adopts the following structural shape: $Y* = X\beta + \varepsilon$. The proportional odds assumption, namely the strength of the association is equal for each category, must be proven to validate this model. After adjusting the standard ordered Logit model, the Brant test failed to verify this assumption [35], therefore showing the inadequate use of this model.

Subsequently, a partial proportional odds model [36], generalisation of the ordered Logit model, was formulated, which relaxes the proportional odds model for a subset of independent variables through different categories of the outcome variable. Its best fit was tested using the Akaike Information Criteria, Bayes Information Criteria, and pseudo-adjusted $R^2$, following the suggestions by Kass and Raftery [37]. Standard errors were estimated using robust methods to account for potential heteroscedasticity [38]. Although this model can be characterised through its coefficients, it can also be interpreted as a nonlinear probability model that allows examining the determinants and the likelihood of occurrence of each outcome, the marginal effects, and particular cases [39]. Therefore, the predictions for certain subgroups of subjects and the marginal effects can be presented, that is, the variability in the probability of expressing a specific category of health status with a change of one unit in the independent variable, assuming the rest of covariates at their mean values. All analyses were performed with the Stata 14.2® software.

## Ethical and legal aspects

Consent for data collection was obtained from participants during the creation of the original data sources by the National Institute of Statistics and the Ministry of Health. In developing health surveys, the Ministry of Health and the National Statistics Institute (NSI) enforce stringent logical, physical, and administrative measures to ensure the effective protection of confidential data from collection to anonymisation. These measures include legal clauses in surveys, formal confidentiality agreements with data collection companies, and secure handling practices. Identifiable information is retained only as long as necessary and is anonymised before publication. ENS microdata files are publicly available and anonymised to prevent any risk of identifying individuals directly or indirectly [30–32]. The authors accessed anonymised data in April 2023. The Ethics Committee for Research from Hospital Universitario Fundación Alcorcón approved the study (ref. 23/62) and the procedures related to data collection.

## Results

The initial sample included 66,168 subjects: 21,007 in 2011, 23,089 in 2017, and 22,072 in 2020. Demographic and health status data were available for all subjects. Information on social class was collected for 63,805 subjects, BMI for 61,996, and marital status for 66,043. Of the total, 66,120 provided details on physical activity, and 66,085 on smoking habits. The average age was 48.24 (SD 18.89, median 53, interquartile range 39–68) years and 51,3% were women. A diagnosis of HTN was found in 5,391 subjects in 2011, 6,244 in 2017, and 6,057 in 2020, resulting in a weighted prevalence of 21.6%. Of those subjects not diagnosed with HTN, 27.3% and 52.2% reported a "very good" and "good" perception of their health status, respectively, versus 6.1% and 41.6% of people with HTN. Table 1 shows the characteristics of the overall sample.

The final model was built with 61,933 observations, incorporating all variables in blocks except nationality, which was not associated with the outcome. Missing values were due to either the subject not providing data or errors during data collection, with no imputation used. The model was adjusted for the survey year, and its pseudo-R2 was 0.2306 (details in Appendix 1). This output can be interpreted as a succession of independent regressions where the coefficients express the association with a worse outcome. First, outcome 1 (very good) is compared versus the remaining possibilities; subsequently, outcomes 1 and 2 (very good and good) are compared against the rest; and so forth. The association between being diagnosed with HTN and the self-perceived health status did not fit the proportional odds model and its association was therefore calculated for each stratum. This association, despite being negative for all the strata, was stronger for the best health status. Something analogous occurred with the association between antihypertensive medication and perception of health status. Appendix 1 presents which variables violated the proportional odds assumption and which did not in the final model. Appendix 2 includes the marginal effects for each explicative variable and shows how a one-unit change in an independent variable affects the likelihood of being in each category of perceived health status, assuming all other variables are at their average values.

Table 2 shows the predicted likelihood of reporting a specific health status as result of model reported in Appendix 1, for the complete sample and particular subgroups, adjusted for the characteristics of each subject. The probability of expressing a "very good" health status was only 14.3% in hypertensive patients with pharmacological treatment versus 17.7% in hypertensive subjects globally and 23.6% in non-hypertensive subjects. The adjusted likelihood of expressing a "bad" or "very bad" health status did not significantly differ between hypertensive subjects, with or without medication prescription, and the general population.

Table 3 displays the percentage differences in the likelihood of reporting a certain health status for different conditions other than HTN. These probabilities stem from the model in Appendix 1 and are adjusted for the rest of the variables at their mean values (Appendix 2 provides details for all the variables in the model). A diagnosis of HTN was associated with a decrease of 6.2 percentage points in the probability of expressing "very good" health while no association was found with "bad" or "very bad" health reports. Taking medication for HTN was also associated with a decrease of 4.5 points in the probability of expressing a "very good" health status.

**Table 1. Characteristics of the study sample. Proportions weighted by sampling method.**

| Survey year | 2011<br>N = 21,007 | 2017<br>N = 23,089 | 2020<br>N = 22,072 | Total<br>N = 66,168 |
|---|---|---|---|---|
| Women | 51.20% | 51.32% | 51.35% | 51.29% |
| Average age (SD) | 47.15 (18.69) | 48.58 (18.88) | 48.95 (19.04) | 48.24 (18.89) |
| Spanish nationality | 88.05% | 90.89% | 89.83% | 89.12% |
| Married | 55.90% | 58.81% | 56.94% | 57.22% |
| Social class | | | | |
| °High | 17.92% | 18.74% | 18.62% | 18.43% |
| °Medium | 32.69% | 32.85% | 32.83% | 32.79% |
| °Low | 49.39% | 48.42% | 48.55% | 48.78% |
| Health status | | | | |
| °Very good | 21.16% | 21.31% | 25.75% | 22.76% |
| °Good | 50.79% | 49.20% | 49.76% | 49.91% |
| °Fair | 20.24% | 21.18% | 17.45% | 19.61% |
| °Bad | 6.27% | 6.38% | 5.52% | 6.06% |
| °Very bad | 1.54% | 1.92% | 1.51% | 1.66% |
| °Smoker | 26.96% | 24.42% | 22.11% | 24.47% |
| °Regular physical activity | 22.46% | 25.88% | 26.60% | 25.00% |
| Weight | | | | |
| °Underweight | 2.37% | 2.50% | 2.29% | 2.39% |
| °Normal weight | 44.97% | 44.46% | 45.28% | 44.90% |
| °Overweight | 36.07% | 36.19% | 36.9% | 36.39% |
| °Obesity | 16.59% | 16.85% | 15.53% | 16.32% |
| Illness-related limitation | | | | |
| °No | 80.25% | 73.99% | 77.33% | 77.18% |
| °Mild | 16.20% | 20.98% | 17.95% | 18.38% |
| °Severe | 3.56% | 5.03% | 4.72% | 4.44% |
| °HTN diagnosis | 21.29% | 22.11% | 21.24% | 21.55% |
| °Other cardiovascular disease | 8.07% | 9.00% | 8.23% | 8.43% |
| °Respiratory disease | 8.47% | 8.91% | 7.64% | 8.33% |
| °Osteoarticular disease | 25.00% | 24.14% | 18.12% | 22.39% |
| °Mental disorder | 11.37% | 13.40% | 11.20% | 12.02% |
| °Current diagnosis of anxiety/depression | 9.74% | 10.45% | 9.03% | 9.73% |
| °Medication for HTN | 16.86% | 19.86% | 18.38% | 18.37% |
| °Polypharmacy | 6.64% | 7.44% | 5.91% | 6.66% |

SD: standard deviation; HTN: arterial hypertension.

Suffering a chronic osteoarticular disease was the clinical condition most strongly associated with a lower probability of reporting a "very good" health status (a decrease of 14.1%), followed by a diagnosis of anxiety or depression in the past year, cardiovascular disease, or chronic respiratory disease (each associated with a decrease of around 9%).

## Discussion

The diagnosis of HTN is associated with a worse self-perception of health status in the general Spanish population. This association reaches significance when reporting the best health status ("very good"), but not with perceptions of "very bad" health status. The medication for HTN shows similar associations but of lesser magnitude, independently of the disease

**Table 2. Mean probability of reporting a specific health status in non-hypertensive and hypertensive patients, with and without related medication, while excluding the effects of other factors. (Appendix 1-2 model).**

| Health status | Very good<br>%<br>(CI95%) | Good<br>%<br>(CI95%) | Fair<br>%<br>(CI95%) | Bad<br>%<br>(CI95%) | Very bad<br>%<br>(CI95%) |
|---|---|---|---|---|---|
| Overall sample | 23.00%<br>(21.26- 24.74%) | 50.55%<br>(49.05- 52.06%) | 19.15%<br>(18.48- 19.81%) | 5.72%<br>(5.31- 6.13%) | 1.58%<br>(1.31- 1.85%) |
| No hypertension | 23.57%<br>(21.76- 25.38%) | 50.91%<br>(49.36- 52.46%) | 18.43%<br>(17.6- 19.27%) | 5.48%<br>(5.05- 5.92%) | 1.60%<br>(1.32- 1.88%) |
| Hypertension | 17.69%<br>(14.88- 20.51%) | 53.07%<br>(49.83- 56.32%) | 21.62%<br>(19.93- 23.31%) | 6.06%<br>(5.47- 6.64%) | 1.56%<br>(1.25- 1.86%) |
| Hypertension with medication | 14.29%<br>(12.55- 16.03%) | 55.42%<br>(53.63- 57.2%) | 22.87%<br>(21.85- 23.89%) | 5.75%<br>(5.2- 6.3%) | 1.67%<br>(1.37- 1.96%) |

CI95%: Confidence interval at 95%

**Table 3. Variability in the probability of reporting a specific perception of health status for different clinical conditions and polypharmacy, adjusted by the remaining characteristics (Appendix 1-2 model).**

| Health status | Very good<br>%<br>(CI95%) | Good<br>%<br>(CI95%) | Fair<br>%<br>(CI95%) | Bad<br>%<br>(CI95%) | Very bad<br>%<br>(CI95%) |
|---|---|---|---|---|---|
| Hypertension | −6.20%<br>(−9.3 to −3.11%) | 2.65%<br>(−1.02 to 6.33%) | 3.03%<br>(1.09–4.97%) | 0.57%<br>(0.02–1.12%) | −0.05%<br>(−0.26 to 0.17%) |
| Cardiovascular disease | −8.99%<br>(−11.93 to −6.06%) | 2.26%<br>(−1.02 to 5.54%) | 4.71%<br>(3.55–5.87%) | 1.35%<br>(0.93–1.77%) | 0.67%<br>(0.46–0.88%) |
| Chronic osteoarticular disease | −14.10%<br>(−17.65 to −10.54%) | 4.55%<br>(1.39–7.7%) | 7.33%<br>(6.5–8.16%) | 1.87%<br>(1.44–2.3%) | 0.34%<br>(0.1–0.59%) |
| Chronic respiratory disease | −8.58%<br>(−9.96 to −7.2%) | 0.79%*<br>(−0.63 to 2.21%) | 5.97%<br>(5.46–6.48%) | 1.48%<br>(1.02–1.95%) | 0.33%<br>(0.16–0.51%) |
| Chronic mental disorder | −7.85%<br>(−9.52 to −6.18%) | 1.73%<br>(1.2–2.26%) | 3.55%<br>(2.74–4.36%) | 1.81%<br>(1.41–2.2%) | 0.76%<br>(0.56–0.95%) |
| Anxiety/depression diagnosis in the last year | −9.50%<br>(−12.22 to −6.78%) | 0.01%<br>(−2.76 to 2.79%) | 7.00%<br>(5.97–8.03%) | 1.83%<br>(1.3–2.36%) | 0.65%<br>(0.15–1.15%) |
| Polypharmacy | −5.39%<br>(−10.91 to 0.14%) | −3.36%<br>(−8.2 to 1.47%) | 6.04%<br>(5.31–6.77%) | 2.07%<br>(1.56–2.59%) | 0.64%<br>(0.39–0.88%) |
| Antihypertensive drugs prescription | −4.52%<br>(−7.29 to −1.75%) | 3.34%<br>(−0.01 to 6.7%) | 1.48%<br>(−0.31 to 3.27%) | −0.50%<br>(−0.9 to −0.11%) | 0.21%<br>(−0.09 to 0.5%) |

CI95%: Confidence interval at 95%

diagnosis. The observed associations were adjusted for those characteristics that were found to modify the perception of health status, such as age, gender, social group, concomitant chronic diseases or limitations, and lifestyles [27,40].

Former studies reported the negative association between HTN and the perception of health status, although without isolating the potential influence of comorbidity or other variables [20,41]. These studies did show that chronic disease was associated with a worse perception of health status in hypertensive patients, but not whether it was differentially so with respect to the general population. The described association has been established in other larger studies. However, whether the decrease in the perceived health status is due to the diagnosis itself or to concern about the disease has not been determined, nor the role of medication [19]. In Spain, a study conducted more than 15 years ago found no association between the diagnosis of HTN and a worse perception of HRQoL nor with the prescription of medication, except in a

subgroup of women who expressed concern about the diagnosis [24]. The referenced study focused on individuals over 60 years of age, with an average age exceeding 70, whereas the current research included a sample where 50% of the subjects were under 53 years old. It has been hypothesized that age is linked to a greater acceptance of chronic conditions, particularly those that are asymptomatic, as health expectations adjust throughout life [27]. Additionally, since only 64.1% of the hypertensive participants in the previous study were aware of their diagnosis, it is possible that the relationship between a hypertension diagnosis and health perception may be influenced by the labeling effect, as suggested by other studies [18].

The findings of the present study also allow estimating the magnitude of the association comparatively. The diagnosis of HTN decreases the probability of expressing a "very good" health status by 6.2 percentage points, a decrease that reaches >14 and > 9 points in the cases of chronic osteoarticular disease or recent diagnosis of anxiety or depression, respectively. The very strong association between osteoarticular pathologies and the perception of HRQoL has been previously documented in European countries [42] and Spain [43], similarly to diagnoses of anxiety or depression [44,45]. Therefore, the impact of the diagnosis of HTN on the perception of the best health status could be comparatively considered significant. In contrast, this diagnosis was not associated with a greater probability of reporting a "bad" or "very bad" health status, as is the case for the other mentioned pathologies.

The intake of medication for HTN in the absence of comorbidities was associated with a decrease of 4.5 percentage points in the probability of expressing a "very good" health status. This is approximately equivalent to 80% of the magnitude of the inverse association between polypharmacy and health status perception, which have been previously reported for older people [46]. Among hypertensive patients, those receiving pharmacological treatment often exhibit a worse perception of their health status compared to those not undergoing such treatment, regardless of blood pressure control [18,19]. Additionally, the impact of hypertension labeling on health perception may, in part, be attributable to the prescription of antihypertensive drugs [21,24]. This study has corroborated the observed trend. However, after adjusting for the presence of other conditions and polypharmacy, it remains uncertain whether the deterioration in health perception is due to awareness of the condition itself, potential side effects of the antihypertensve medication, or other related factors.

Both the diagnosis of HTN and the prescribed medication appear to be independently associated with a less positive health perception. This study did not assess whether better control of the disease improves this perception, or whether this association is affected by differences in coping with the disease, as proposed by other authors [14,15]. The present findings point to confirm that the referred associations were not solely related to the existence of comorbidity or ageing and that the perception of health in hypertensive patients worsens even if their general physical condition is not affected [41]. Whether the worse perception of health status is due to concern about the disease, blood pressure values, or the recommended lifestyle changes could not be established. However, a conclusion could be reached that this impact was not due to disease-related complications and that its magnitude was moderate compared to other conditions, as shown in other research [19].

The implications of these findings are significant, offering a deeper understanding of how hypertension affects health perception in Spain. This study considers a broad range of factors, such as medication, comorbidities, and sociodemographic characteristics, providing valuable insights into how hypertension compares to other chronic conditions. These findings are crucial for health professionals to understand the potential impact of a hypertension diagnosis on patients' health perception and to inform the development of more effective intervention strategies aimed at improving patients' coping mechanisms and adherence to treatment.

Additionally, the study confirms that the negative perception of health in hypertensive patients is not solely linked to comorbidities or aging. Although we did not evaluate the effect of blood pressure control on health perception, our results show that both the diagnosis of hypertension and prescribed medication independently contribute to a less positive health perception. This suggests that while awareness of hypertension may enhance treatment adherence [12,15], it may also lead to a worsened perception of health. Understanding this impact, independent of disease complications, is

vital for improving patient outcomes and guiding healthcare interventions. While disease control does not seem to affect perceived health status [23,24], medication use is associated with a decline in health perception. In community-based settings, where healthcare providers have more direct interactions with patients, this understanding is especially important. It enables healthcare professionals to tailor interventions that not only address the clinical aspects of hypertension but also focus on improving patients' overall well-being, coping mechanisms, and health perception. Strategies should aim to enhance both treatment adherence and psychological coping with the condition, considering that hypertension is a major cardiovascular risk factor. Additionally, socioeconomic status should be taken into account, as it likely influences how individuals manage their condition. These insights are critical for health planning and policy development, emphasizing the need to integrate patients' perspectives into healthcare practices to better address their needs and improve their health perception.

Among the study's limitations is its cross-sectional design, which allows for assessing associations but not causality. Additionally, all data were self-reported, subject to reporting biases, and the lack of blood pressure data prevented classification of diagnoses into different stages. However, a strength of the study is the collection of data over an extended period from the most representative sample of the Spanish population. The survey methodology, which involved home visits, may have led to an underrepresentation of socially excluded groups. Despite these limitations, the findings remain highly relevant, providing a nuanced understanding of how hypertension affects health perception.

In conclusion, this research established the association between the diagnosis of HTN and a lower probability of expressing perceptions of "very good" health status at the population level. Such association was not mediated by the presence of other comorbidities or related complications. Medication for HTN was also found to be independently associated with expressions of less satisfactory health status.

## Supporting information

**Appendix 1. Final explanative model.**
(DOCX)

**Appendix 2. Final explanative model. Marginal effects.**
(DOCX)

**RECORD Checklist_rev2.**
(DOCX)

**Dataset_Plos_weighting_def.**
(XLSX)

## Author contributions

**Conceptualization:** Jesús Martín-Fernández, Tamara Alonso-Safont, Elena Polentinos-Castro, Gemma Rodríguez-Martínez, Mª Isabel González-Anglada, Amaia Bilbao-González, Isabel del Cura-González.

**Data curation:** Jesús Martín-Fernández, Elena Polentinos-Castro, Isabel del Cura-González.

**Formal analysis:** Jesús Martín-Fernández, Tamara Alonso-Safont, Amaia Bilbao-González.

**Funding acquisition:** Jesús Martín-Fernández, Amaia Bilbao-González.

**Investigation:** Jesús Martín-Fernández, Tamara Alonso-Safont, Elena Polentinos-Castro, Gemma Rodríguez-Martínez, Mª Isabel González-Anglada, Amaia Bilbao-González, Isabel del Cura-González.

**Methodology:** Jesús Martín-Fernández, Tamara Alonso-Safont, Elena Polentinos-Castro, Gemma Rodríguez-Martínez, Mª Isabel González-Anglada, Amaia Bilbao-González, Isabel del Cura-González.

**Project administration:** Jesús Martín-Fernández.

**Resources:** Jesús Martín-Fernández.

**Supervision:** Isabel del Cura-González.

**Validation:** Jesús Martín-Fernández, Tamara Alonso-Safont, Elena Polentinos-Castro, Gemma Rodríguez-Martínez, Mª Isabel González-Anglada, Amaia Bilbao-González, Isabel del Cura-González.

**Writing – original draft:** Jesús Martín-Fernández.

**Writing – review & editing:** Jesús Martín-Fernández, Tamara Alonso-Safont, Elena Polentinos-Castro, Gemma Rodríguez-Martínez, Mª Isabel González-Anglada, Amaia Bilbao-González, Isabel del Cura-González.

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
