## [Decision Letter · Decision Letter 0]

7 Aug 2024

PONE-D-23-29484"Does a diagnosis of arterial hypertension affect the perception of health status? A decade-long study in the Spanish population"PLOS ONE

Dear Dr. Martín-Fernández,

Thank you for submitting your manuscript to PLOS ONE. After careful consideration, we feel that it has merit but does not fully meet PLOS ONE’s publication criteria as it currently stands. Therefore, we invite you to submit a revised version of the manuscript that addresses the points raised during the review process.

**ACADEMIC EDITOR: **Both reviewers felt that the novelty of this work is rather limited. 

Specifically, a number of factors, including the level of literacy or level of education of the population, and to some extent the socioeconomic status of the population, should be elaborated to highlight the impact and novelty of the study.

The results and interpretation of the model should be revised for better clarification. 

We look forward to receiving your revised manuscript.

Kind regards,

Hoh Boon-Peng, PhD

Academic Editor

PLOS ONE

Reviewers' comments:

Reviewer's Responses to Questions

**Comments to the Author**

1. Is the manuscript technically sound, and do the data support the conclusions?

Reviewer #1: Yes

Reviewer #2: No

2. Has the statistical analysis been performed appropriately and rigorously? 

Reviewer #1: Yes

Reviewer #2: No

3. Have the authors made all data underlying the findings in their manuscript fully available?

Reviewer #1: No

Reviewer #2: No

4. Is the manuscript presented in an intelligible fashion and written in standard English?

Reviewer #1: Yes

Reviewer #2: Yes

5. Review Comments to the Author

Reviewer #1: Reviewer’s report

(Manuscript ID: PONE-D-23-29484)

Thank you very much for asking me to review the manuscript titled ‘Does a diagnosis of arterial hypertension affect the perception of health status? A one decade-long study in the Spanish population’. My comments are appended below

General Comment

This cross-sectional study assessed the impact of the diagnosis of hypertension on self-perceived health status, and attempted to identify some probable factors that may influence this perception. The sources of the data were the National Health Surveys of 2011-2012, 2017 and the 2020 European Health Interview Survey for Spain. From this a total of 66,168 subjects were included. Stratified sampling was performed to get the required subjects from each of these surveys. The statistical analyses that was performed seems adequate. Following the analyses, the researchers concluded that the diagnosis of hypertension and prescription of antihypertensive medication are associated with a lower probability of reporting a “very good” health status, independent of other comorbidities or complications related to the diagnosis. No other factors, including age, were found associated with the perception. Whilst the findings are interesting, they are however not really that novel and do not really add any substantial new information to what we already know. Decreased perception of well-being can be expected with any disease, which may depend on a number of factors, including the level of literacy or level of education of the population, and to some extent the socioeconomic status of the population.

Specific comments

1. Although the sample size is indeed large, no information on sample size calculation has been included.

2. They had indicated the fraction of the population that was considered ‘high class’, but no information was given on how this was derived? Was it based on their income or the locality where they lived or?????

3. What fraction was middle class or lower middle class etc?

4. No information has been given on the educational status of the study population. This is important as is most likely going to impact the self-perception. It would have been good if information on the educational status of the study population had been included.

5. Please include a table showing the results of statistical analyses of the associations between HTN and age and other co-morbidities.

6. It was interesting to note that those on medication for hypertension were less likely to score a ‘very good’ perception than those who were not taking any medication but knew they had hypertension. What could be the reason for this? No discussion was attempted on this.

7. Why was medication found to be independently associated with the expression of less satisfactory health status? Any speculation?

8. The discussion could be improved in other places too, like e.g. they state that their findings were different from those reported in another survey on the population 15 years earlier but provide no reason or speculation as to why they are different. What was/were the difference/s between these two studies?

9. They state that there was a total of 66,168 participants, but in Table 1 the total number for smokers and non-smokers is 66085; for body weight it is 61996. What happened to the missing subjects? Please ensure the totals tally and if they don’t then provide a reason.

10. They need to emphasise a little more on the relevance and importance of these findings in the discussion. How this will influence the management of hypertension? We know compliance to treatment is one of the major problems in the management of hypertension. Does the perception of the illness impact compliance?

11. The title could be revised to be a little more informative

Reviewer #2: The manuscript titled "Does a diagnosis of arterial hypertension affect the perception of health status? A decade-long study in the Spanish population" is an interesting study, conducted in a community setting, that aimed to assess and discuss the association of being diagnosed with arterial hypertension with self-perceived health status and potential explanative factors.

However, the research gap or the novelty of the manuscript is inadequately highlighted, as previous studies had evaluated effects of arterial hypertension on health-related quality of life. This study employed a generalised ordered logit model in its statistical analysis. However, the results and interpretation of the model are not clearly described. In addition, evidence of statistical significance of some of the results described in the manuscript is not available. As a consequence, the discussion and conclusion can only be reviewed objectively and constructively after the authors have addressed the related comments accordingly.

Please find my detailed comments in the attachment.

6. PLOS authors have the option to publish the peer review history of their article (what does this mean? ). If published, this will include your full peer review and any attached files.

**Do you want your identity to be public for this peer review?** For information about this choice, including consent withdrawal, please see our Privacy Policy .

Reviewer #1: No

Reviewer #2: No

---

## [Author Response · Author response to Decision Letter 1]

30 Aug 2024

To the Editorial Team:

We sincerely appreciate the thorough review of our manuscript now entitled, following the suggestion of Reviewer 1 “How a diagnosis of arterial hypertension influences health status perception: findings from a decade-long study in the Spanish population” We are sincerely grateful for the efforts of the editors and reviewers, whose work has substantially enhanced the manuscript. We are confident that these improvements will contribute significantly to the final outcome.

First of all, we have slightly modified the wording of the objective to better align it with the idea we wanted to convey. Its wording is now as follows: “This study, conducted in the community setting, aimed to assess and discuss how a diagnosis of arterial hypertension affects self-perceived health status, examining the association with potential explanatory factors and comparing its impact with that of other chronic conditions”.

Additionally, we would like to note that the authors have recognized the potential for improvement in the data analysis during the review process. The data comes from three national surveys in which sampling was conducted using clusters to ensure representativeness at the Autonomous Community level. This means that each subject has a different level of representativeness, which is accounted for in a weighting factor. Consequently, all analyses have been corrected to take this weighting factor into account. As a result, the findings in Table 1 and the model (Appendix 1, Appendix 2 and Tables 2 and 3) differ slightly from the original results, although the direction and magnitude of the reported relationships remain unchanged.

We will now proceed to address the suggestions and corrections proposed by Editor and reviewers.

ACADEMIC EDITOR:

Both reviewers felt that the novelty of this work is rather limited.

Specifically, a number of factors, including the level of literacy or level of education of the population, and to some extent the socioeconomic status of the population, should be elaborated to highlight the impact and novelty of the study.

The results and interpretation of the model should be revised for better clarification.

We appreciate the opportunity to highlight the novelty of this study. As we detail further (see response to the reviewers), it was not clear whether a diagnosis of hypertension (HTN) affects the perception of health status when there are no cardiovascular complications. Some studies in our country question this issue. Additionally, studies that claim this impact exists are unsure whether to attribute it to labeling effects, medication, potential side effects of the medication, or certain complications. In this work, we clarify some, though not all, of these aspects. We found that the diagnosis already affects the perception of health status, regardless of whether medication is prescribed and without being due to potential complications of hypertension.

We agree on the importance of adjusting health status perception for socioeconomic characteristics such as education level or social class. In this study, a classification of social class based on occupation was used as an adjustment variable, which includes professionals with higher education in the “higher social class” group. We explain these details and their implications in our response to the reviewers below.

Regarding the results and interpretation of the model, we have added the results in a new appendix in their most interpretable form, as marginal effects. These results show how the probability of expressing a certain category (perception of health status) changes when the explanatory variable is modified, while keeping the other variables constant at their means (we present the marginal effects at the means). It is worth noting that one of the reviewers mentioned being unable to evaluate the model due to not having access to it. This model was sent as an annex, but it seems the reviewer was unable to access it.

Reviewer #1: Reviewer’s report

General Comment

This cross-sectional study assessed the impact of the diagnosis of hypertension on self-perceived health status, and attempted to identify some probable factors that may influence this perception. The sources of the data were the National Health Surveys of 2011-2012, 2017 and the 2020 European Health Interview Survey for Spain. From this a total of 66,168 subjects were included. Stratified sampling was performed to get the required subjects from each of these surveys. The statistical analyses that was performed seems adequate. Following the analyses, the researchers concluded that the diagnosis of hypertension and prescription of antihypertensive medication are associated with a lower probability of reporting a “very good” health status, independent of other comorbidities or complications related to the diagnosis. No other factors, including age, were found associated with the perception. Whilst the findings are interesting, they are however not really that novel and do not really add any substantial new information to what we already know. Decreased perception of well-being can be expected with any disease, which may depend on a number of factors, including the level of literacy or level of education of the population, and to some extent the socioeconomic status of the population.

We sincerely appreciate the comments provided. The impact of a hypertension diagnosis on health status remains unclear. While some studies report a significant impact, others suggest that it only occurs when medication is introduced (Trevisol DJ, et al. Health-related quality of life is worse in individuals with hypertension under drug treatment: results of population-based study. J Hum Hypertens. 2012) or when cardiovascular diseases develop (Alefishat E, et al Factors affecting health‐related quality of life among hypertensive patients using the EQ‐5D tool. Int J Clin Pract. 2020). In fact, it has been noted that, as hypertension is an asymptomatic condition, patients often do not perceive it as a disease and may not adhere to medical recommendations (Katsi V, et al. Arterial hypertension and health-related quality of life. Front Psychiatry. 2017). We have added information to the introduction section, noting that two studies in our country have reported that, in individuals over 60 years of age, a hypertension diagnosis was not associated with the perception of health status (Banegas J, et al. Association Between Awareness, Treatment, and Control of Hypertension, and Quality of Life Among Older Adults in Spain. Am J Hypertens. 2006; Forjaz MJ, et al Chronic conditions, disability, and quality of life in older adults with multimorbidity in Spain. Eur J Intern Med. 2015). The importance of our study lies in demonstrating that the mere diagnosis of hypertension, regardless of whether medication is prescribed or complications arise, affects health perception. We believe that the findings from this representative sample of the Spanish population may shed some light on the ongoing debate.

Specific comments

1. Although the sample size is indeed large, no information on sample size calculation has been included.

The sample size in each survey wave was determined by its design, estimated to achieve representativeness at the Autonomous Community level. This information, along with the selection criteria and sampling methodology, has been included in the Methods section.

Here’s the modified paragraph:

“Stratified sampling was performed in three phases as part of the survey methodology: first, census sections were selected (2,000 sections in 2011 and 2,500 sections in 2017 and 2020); second, 12 households were chosen from each section in 2011, and 15 households from each section in 2017 and 2020; third, one adult aged 15 years or older was selected from each household. The only selection criteria were being 15 years or older and living in the selected household. All individuals meeting this condition had an equal probability of being selected, but only one person was included from each household. The sample size for each survey was calculated to ensure representativeness at the regional level (Autonomous Community) (29–31)”.

2. They had indicated the fraction of the population that was considered ‘high class’, but no information was given on how this was derived? Was it based on their income or the locality where they lived or?????

The classification of social class used in this study is based on a widely recognized proposal that categorizes individuals based on occupation, which is highly correlated with educational background. This proposal includes six categories. Categories I and II, referred to as "High social class" in this study, encompass directors and managers of business establishments, regardless of whether they work in public administration or private enterprise; professionals with university-level education; as well as athletes and artists. Details of the classification can be found in reference 32 (Domingo-Salvany A, et al. Propuestas de clase social neoweberiana y neomarxista a partir de la Clasificación Nacional de Ocupaciones 2011. Gac Sanit. 2013). In the Methods section, we explain the occupational categories included in the so-called "High classes."

3. What fraction was middle class or lower middle class etc?

We provide a more detailed breakdown of social class in 6 groups for this dataset below, although we believe it may not be necessary in the manuscript as it is merely an adjustment variable. If the editors or reviewers feel otherwise, we have no objection to include the information.

See table in attached file.

4. No information has been given on the educational status of the study population. This is important as is most likely going to impact the self-perception. It would have been good if information on the educational status of the study population had been included.

We agree with the reviewer on the importance of including social characteristics, such as educational level, in health perception studies. It should be noted that in the 2011-12 National Health Survey, the educational level variable was collected for the entire household, not individually, as is the case in the two subsequent surveys. However, our classification of social class, based on occupation, is highly correlated with education. Categories I and II, labeled "High social class," include directors, managers, university-educated professionals, athletes, and artists. All professionals with high education are encompassed within these categories. Given the strong correlation between occupation and educational level, these two variables are typically not used together in models to avoid high collinearity. Therefore, we have included the description of "High social class" in the Methods section to address this issue.

5. Please include a table showing the results of statistical analyses of the associations between HTN and age and other co-morbidities.

We are not sure what the reviewer's concern is at this point. The age distribution of hypertension in the Spanish population is well documented in many sources (see, for example, Menéndez E et al . Prevalencia, diagnóstico, tratamiento y control de la hipertensión arterial en España. Resultados del estudio Di@ bet. es. Revista española de cardiología, 2016). Below, we present the percentages of hypertensive individuals within the disease groups studied.

See table in attached file.

In this other table, the percentages of different conditions within the hypertensive and non-hypertensive groups are shown.

See table in attached file.

We are unsure if this is of interest given the study's objectives. Perhaps the reviewer is referring to the need to show the relationships between the adjustment variables and the perception of health status (age, gender, limitation, lifestyles, etc.). We have included all the results of the model in its more interpretable form, the marginal effects, in Appendix 2. Table 3 is a summary of this appendix, featuring variables that not only serve as adjustment variables but also allow for comparing the magnitude of the impact of hypertension on the perception of health status, as proposed in the objective. This way, readers will have all the information available in the appendix as well as the necessary details to follow the main text of the manuscript.

6. It was interesting to note that those on medication for hypertension were less likely to score a ‘very good’ perception than those who were not taking any medication but knew they had hypertension. What could be the reason for this? No discussion was attempted on this.

The concept of treatment burden, has been extensively documented in the literature. It refers to the overall workload imposed on patients due to their healthcare regimen, which often results in various negative impacts on their lives. Research has also established a link between this burden and medication adherence (Zidan A, et al. Medication-Related Burden among Patients with Chronic Disease Conditions: Perspectives of Patients Attending Non-Communicable Disease Clinics in a Primary Healthcare Setting in Qatar. Pharmacy (Basel). 2018). There are several reasons why a person might have a worse perception of their health status upon receiving a prescription. We can only hypothesize about these reasons. It has been described that inappropriate prescriptions or polypharmacy can worsen the perception of quality of life (Olsson IN et al . Medication quality and quality of life in the elderly, a cohort study. Health Qual Life Outcomes 2011). In the case of hypertensive patients, it has already been noted that those receiving pharmacological treatment had a worse perception of their health status compared to those who did not receive such treatment (Trevisol DJ, et al. Health-related quality of life is worse in individuals with hypertension under drug treatment: results of population-based study Journal of Human Hypertension 2011). However, it cannot be determined whether this was due to concern about the illness or potential side effects of the medication. We include these ideas in the Discussion section

7. Why was medication found to be independently associated with the expression of less satisfactory health status? Any speculation?

Thank you for your suggestion. We have addressed issues 6 and 7 by incorporating the following paragraph:

Among hypertensive patients, those receiving pharmacological treatment often exhibit a worse perception of their health status compared to those not undergoing such treatment, regardless of blood pressure control (20,40). Additionally, the impact of hypertension labeling on health perception may, in part, be attributable to the prescription of antihypertensive drugs (24). This study has corroborated the observed trend. However, after adjusting for the presence of other conditions and polypharmacy, it remains uncertain whether the deterioration in health perception is due to awareness of the condition itself, potential side effects of the antihypertensve medication, or other related factors.

8. The discussion could be improved in other places too, like e.g. they state that their findings were different from those reported in another survey on the population 15 years earlier but provide no reason or speculation as to why they are different. What was/were the difference/s between these two studies?

You are right. We have added the following text:

The referenced study focused on individuals over 60 years of age, with an average age exceeding 70, whereas the current research included a sample where 50% of the subjects were under 53 years old. It has been hypothesized that age is linked to a greater acceptance of chronic conditions, particularly those that are asymptomatic, as health expectations adjust throughout life (26). Additionally, since only 64.1% of the hypertensive participants in the previous study were aware of their diagnosis, it is possible that the relationship between a hypertension diagnosis and health perception may be influenced by the labeling effect, as suggested by other studies (40).

9. They state that there was a total of 66,168 participants, but in Table 1 the total number for smokers and non-smokers is 66085; for body weight it is 61996. What happened t

---

## [Decision Letter · Decision Letter 1]

20 Dec 2024

PONE-D-23-29484R1How a diagnosis of arterial hypertension influences health status perception: findings from a decade-long study in the Spanish population.PLOS ONE

Dear Dr. Martín-Fernández,

Thank you for submitting your manuscript to PLOS ONE. After careful consideration, we feel that it has merit but does not fully meet PLOS ONE’s publication criteria as it currently stands. Therefore, we invite you to submit a revised version of the manuscript that addresses the points raised during the review process.

**ACADEMIC EDITOR: **Although one reviewer has recommended accept for publication. However, another reviewer, which has the same opinion with Reviewer#2, raised concern by the lack of novelty of this work, as there are multiple works similar in nature being published previously. As much as the work is scientifically valid, what are the new knowledge / information that would fill the current research gap is essential. I suggest the authors to relook this matter, and highlight the novelty / new knowledge that would narrow the research gap. ==============================

We look forward to receiving your revised manuscript.

Kind regards,

Hoh Boon-Peng, PhD

Academic Editor

PLOS ONE

Additional Editor Comments:

Although one reviewer has recommended accept for publication. However, another reviewer, which has the same opinion with Reviewer#2, raised concern by the lack of novelty of this work, as there are multiple works similar in nature being published previously. As much as the work is scientifically valid, what are the new knowledge / information that would fill the current research gap is essential. I suggest the authors to relook this matter, and highlight the novelty / new knowledge that would narrow the research gap.

Reviewers' comments:

Reviewer's Responses to Questions

**Comments to the Author**

1. If the authors have adequately addressed your comments raised in a previous round of review and you feel that this manuscript is now acceptable for publication, you may indicate that here to bypass the “Comments to the Author” section, enter your conflict of interest statement in the “Confidential to Editor” section, and submit your "Accept" recommendation.

Reviewer #1: All comments have been addressed

Reviewer #3: All comments have been addressed

2. Is the manuscript technically sound, and do the data support the conclusions?

Reviewer #1: Yes

Reviewer #3: Partly

3. Has the statistical analysis been performed appropriately and rigorously? 

Reviewer #1: Yes

Reviewer #3: Yes

4. Have the authors made all data underlying the findings in their manuscript fully available?

Reviewer #1: Yes

Reviewer #3: Yes

5. Is the manuscript presented in an intelligible fashion and written in standard English?

Reviewer #1: Yes

Reviewer #3: Yes

6. Review Comments to the Author

Reviewer #1: Dear Editor,

Thank you very much for asking me to review the revised manuscript titled ‘How a diagnosis of arterial hypertension influences health status perception: findings from a one decade-long study in the Spanish population’. My comments are appended below

Overall Comment

The authors have made a substantial effort in addressing the reviewers’ comments. I am satisfied with the revisions and their rebuttal. However, you may want to re-look at the following before accepting it for publication.

Minor comments

1. It will be useful to the readership if the fractions of middle or lower middle class of the study population are also included.

2. I still think it is only appropriate for them to just state some of the reasons for the differences in the number of participants in the initial and final analysis. I realise there are always going to be incomplete data entries by participants when dealing with population surveys, but it is, nevertheless, appropriate to state the reasons briefly, instead of expecting the readers to refer to the database for reasons of the differences in the initial and final study population.

3. With reference to the title, it could be revised further in my opinion. The study data presented is not indicating about the ’how’ or the ‘why’ the perception of health is the way it is. It is rather documenting an association between the diagnosis and treatment of hypertension and the probability of negative perception of their health status. Now, it is not wrong to have a title that clearly includes the main message arising from the study itself. They may, therefore, wish to consider the following title - “Diagnosis of hypertension and antihypertensive prescription is associated with a negative self-perception of health status”. It is what they have concluded from the study.

4. The word ‘data’ is a plural word, and they will need to revise the words ‘data was’ to ‘data were’ in a couple of places in the manuscript.

Reviewer #3: There have been several previous reports on the perception of health among patients with hypertension which therefore lacks the novelty of this report.

7. PLOS authors have the option to publish the peer review history of their article (what does this mean? ). If published, this will include your full peer review and any attached files.

**Do you want your identity to be public for this peer review?** For information about this choice, including consent withdrawal, please see our Privacy Policy .

Reviewer #1: No

Reviewer #3: No

---

## [Author Response · Author response to Decision Letter 2]

28 Dec 2024

To the Editorial Team:

We sincerely appreciate the opportunity to revise and resubmit our manuscript, formerly titled “How a diagnosis of arterial hypertension influences health status perception: findings from a decade-long study in the Spanish population,” as well as the dedication of the editorial team and reviewers in providing valuable feedback to improve our work.

We understand the concern raised regarding the perceived lack of novelty and would like to emphasize the unique contribution of our study. Our research is based on data from a nationally representative survey with a robust methodological design, conducted over more than ten years. The scale and representativeness of this dataset provide a comprehensive and reliable foundation to explore how the diagnosis of hypertension impacts health status perception. These results offer valuable insights that are directly applicable to public health policy and clinical practice, addressing a gap in the understanding of this relationship within large and diverse populations.

We are committed to addressing the concerns about novelty and clearly highlighting the knowledge our study adds to fill existing research gaps in the revised manuscript. We are confident that these enhancements will further enrich the manuscript.

ACADEMIC EDITOR:

Although one reviewer has recommended accept for publication. However, another reviewer, which has the same opinion with Reviewer#2, raised concern by the lack of novelty of this work, as there are multiple works similar in nature being published previously. As much as the work is scientifically valid, what are the new knowledge / information that would fill the current research gap is essential. I suggest the authors to relook this matter, and highlight the novelty / new knowledge that would narrow the research gap.

We acknowledge that the relationship between hypertension and health perception has been explored in diverse contexts and through various methods. However, this link remains not fully understood, with studies often producing contradictory findings.

In 2008 a study analyzing HRQoL among 8,303 adults from the 2001–2004 NHANES cohort found that individuals with hypertension reported poorer health outcomes, including fair or poor health status, more unhealthy days, and increased activity limitations. Those aware of their hypertension had even worse HRQoL, with more physically unhealthy days and mental health challenges. Antihypertensive medication was linked to more physically unhealthy days, but HRQoL did not differ significantly based on blood pressure control status (Hayes et al., 2008). However, the study did not examine the impact of comorbidities on these outcomes.

In 2011, Trevisol et al., in a systematic review of over 20 studies, concluded that the health perception of individuals with hypertension is slightly worse than that of normotensive individuals. The review emphasized that the influence of high blood pressure and the awareness of having hypertension require further investigation (Trevisol et al., 2011).

One year later, the same author conducted a population-based study with nearly 2,000 patients and concluded that the worse health perception among individuals with hypertension is more evident in patients undergoing treatment, particularly when their blood pressure is controlled by medication (Trevisol et al., 2012).

Other studies have explored the role of comorbidities in reducing health perception attributed to hypertension, finding that hypertension itself significantly reduces health perception. This impact is further influenced by various factors, such as gender, education level, blood pressure control, and physical exercise. Numerous studies have also highlighted a strong association between drug-related factors—such as the number of prescribed medications, disease duration and severity, and the presence of adverse drug reactions or complications—and diminished health perception (Shah et al., 2020).

In contrast, Katsi et al. (Katsi et al., 2017), in an observational study of a European population, concluded that the stage of hypertension and awareness of the disease do not significantly affect patients' physical or mental health. This finding suggests that the asymptomatic nature of hypertension in its early stages may explain the lack of impact on health perception.

Moreover, some studies have shown that health perception in hypertensive patients can differ based on personal, regional, and contextual factors. For example, differences in health perception have been observed between urban and rural populations, with factors such as education, marital status, employment, health behaviors, and access to healthcare influencing these differences (Zhang et al., 2016). This highlights the importance of considering both individual and contextual factors when investigating health status perception.

Therefore, it is particularly relevant to examine the impact of hypertension within the specific social and cultural context of Spain. Understanding how the disease is perceived and managed in this unique context is essential for developing targeted interventions that address the particular needs and circumstances of the population. This approach will provide a more accurate and culturally sensitive understanding of how hypertension influences health perception.

In evaluating the published evidence in Spain, it becomes evident that the relationship between hypertension and health perception is complex and not fully understood. While hypertension is a well-established risk factor for various health complications, its direct impact on individuals' perception of their health appears limited when it occurs without comorbidities.

For instance, a study conducted in Spain in 2006, involving 3,368 participants aged 60 or older, used the SF-36 questionnaire to assess HRQL. The study found no significant association between hypertension, antihypertensive treatment, or blood pressure control and worse HRQL in the overall sample. These findings suggest that, in the absence of additional health conditions, hypertension does not directly deteriorate health perception (Banegas et al., 2006).

Similarly, Forjaz et al. analyzed chronic conditions, disability, and quality of life among older adults with multimorbidity in Spain. They concluded that, although hypertension was highly prevalent, it was not significantly associated with worse quality of life (Forjaz et al., 2015).

Further supporting this perspective, a population based study of over 11,000 individuals examined the joint impact of nutritional status and chronic diseases on self-rated health. Among participants with normal weight, hypertension alone was not associated with a higher probability of reporting poor health status. However, when hypertension coexisted with overweight or obesity, the likelihood of reporting poor health status increased substantially, particularly for those with additional chronic conditions such as diabetes (Sayón-Orea et al., 2018). These results reinforce the idea that hypertension in isolation has a limited impact on health perception, whereas the combination of hypertension with other health challenges significantly amplifies the likelihood of reporting poorer health.

In light of these findings, we argue that the relationship between hypertension and health perception cannot be generalized across all populations and contexts. It is essential to consider the broader social, cultural, and individual factors that influence this relationship. The present study seeks to evaluate the impact of a hypertension diagnosis on health perception, while carefully isolating the effects of medication, comorbidities, and other circumstances known to influence this perception. This approach addresses critical gaps in understanding how the diagnosis itself shapes health perception independently of confounding factors.

Furthermore, our study compares the impact of hypertension on health perception with that of other chronic diseases, which provides valuable context for understanding the magnitude of hypertension's impact on health perception. By leveraging the most representative population health data source available in our context, constructed with a robust and reliable methodology, the study ensures that the findings are grounded in high-quality evidence. This makes the findings highly relevant to clinical and public health discussions.

By addressing these gaps, the study contributes to a nuanced understanding of how hypertension affects individuals' health perception, laying the groundwork for tailored interventions aimed at improving outcomes in hypertensive patients. We believe this justifies the pertinence of conducting this research.

Some sentences in the introduction have been rewritten to reinforce the pertinence of the study (see the version with tracked changes).

The section on the implications of the study has been rewritten to highlight the contribution of this work to the current state of knowledge.

Reviewer #1: Reviewer’s report

Thank you very much for asking me to review the revised manuscript titled ‘How a diagnosis of arterial hypertension influences health status perception: findings from a one decade-long study in the Spanish population’. My comments are appended below

Overall Comment

The authors have made a substantial effort in addressing the reviewers’ comments. I am satisfied with the revisions and their rebuttal. However, you may want to re-look at the following before accepting it for publication.

Thank you for your feedback. We appreciate your comments and will review the remaining points carefully.

Minor comments

1. It will be useful to the readership if the fractions of middle or lower middle class of the study population are also included.

The weighted percentages of middle and lower social classes have been incorporated into Table 1.

2. I still think it is only appropriate for them to just state some of the reasons for the differences in the number of participants in the initial and final analysis. I realise there are always going to be incomplete data entries by participants when dealing with population surveys, but it is, nevertheless, appropriate to state the reasons briefly, instead of expecting the readers to refer to the database for reasons of the differences in the initial and final study population.

The final model was built with 61,933 observations, representing the number of subjects for whom all variables included in the model were available. Missing values resulted from either the interviewed subject not providing the data or losses and errors during data collection and management. No imputation method was used for missing data.

The information provided in the next paragraph is included in the Results section to address your suggestion. This ensures that readers can understand the reasons for the differences in the number of participants between the initial and final analysis without needing to refer to the database.:

“The initial sample included 66,168 subjects: 21,007 in 2011, 23,089 in 2017, and 22,072 in 2020. Demographic and health status data were available for all subjects. Information on social class was collected for 63,805 subjects, BMI for 61,996, and marital status for 66,043. Of the total, 66,120 provided details on physical activity, and 66,085 on smoking habits”.

3. With reference to the title, it could be revised further in my opinion. The study data presented is not indicating about the ’how’ or the ‘why’ the perception of health is the way it is. It is rather documenting an association between the diagnosis and treatment of hypertension and the probability of negative perception of their health status. Now, it is not wrong to have a title that clearly includes the main message arising from the study itself. They may, therefore, wish to consider the following title - “Diagnosis of hypertension and antihypertensive prescription is associated with a negative self-perception of health status”. It is what they have concluded from the study.

We appreciate your suggestion and have carefully reviewed the title in light of your feedback. However, after consideration, we are not fully convinced that incorporating the study's main result directly into the title is the best approach. Instead, we believe a broader title better captures the scope of the research. Following your input, we have decided on the following title: " Association between hypertension and self-perception of health status: Findings from a decade population-based survey in Spanish adults "

4. The word ‘data’ is a plural word, and they will need to revise the words ‘data was’ to ‘data were’ in a couple of places in the manuscript.

Thank you for your suggestion. We have made the correction.

Reviewer #3: Reviewer’s report

There have been several previous reports on the perception of health among patients with hypertension which therefore lacks the novelty of this report.

We appreciate the reviewer's comment, but as mentioned in our response to the editor, we hold a different opinion. While we acknowledge that the relationship between hypertension and health perception has been explored in various contexts, it remains incompletely understood, with many studies yielding contradictory results. For instance, research has shown that hypertension alone may not always lead to worse health perception, particularly when no comorbidities are present. Furthermore, studies have indicated that factors such as gender, education, medication use, and comorbid conditions can significantly influence how individuals perceive their health. However, research on this topic remains fragmented, with varying conclusions based on the context and population studied. Our study aims to address these gaps, offering a more comprehensive understanding of the impact of hypertension diagnosis on health perception within the specific social and cultural context of Spain. This focused approach, isolating medication, comorbidities, and other relevant factors, is intended to provide clearer insights into how hypertension itself influences health perception. We believe our study fills an important gap in the existing literature and justifies the relevance of this research.

Both the introduction and the discussion have been updated with new paragraphs to emphasize the contribution of this study to the knowledge on the relationship between hypertension and health perception in our context.

We greatly appreciate all the comments and suggestions provided, as we believe they will contribute to improving the quality of the manuscript. We hope that this revised version meets the necessary standards for publication, and we remain open to addressing any additional points you feel may require further revision.

Sincerely,

Dr. Jesús Martín-Fernández

On behalf of the authors

---

## [Decision Letter · Decision Letter 2]

24 Mar 2025

Association between hypertension and self-perception of health status: Findings from a decade population-based survey in Spanish adults.

PONE-D-23-29484R2

Dear Dr. Martín-Fernández,

We’re pleased to inform you that your manuscript has been judged scientifically suitable for publication and will be formally accepted for publication once it meets all outstanding technical requirements.

Kind regards,

Hoh Boon-Peng, PhD

Academic Editor

PLOS ONE

Additional Editor Comments (optional):

Reviewers' comments:

Reviewer's Responses to Questions

**Comments to the Author**

1. If the authors have adequately addressed your comments raised in a previous round of review and you feel that this manuscript is now acceptable for publication, you may indicate that here to bypass the “Comments to the Author” section, enter your conflict of interest statement in the “Confidential to Editor” section, and submit your "Accept" recommendation.

Reviewer #3: All comments have been addressed

2. Is the manuscript technically sound, and do the data support the conclusions?

Reviewer #3: Yes

3. Has the statistical analysis been performed appropriately and rigorously? 

Reviewer #3: Yes

4. Have the authors made all data underlying the findings in their manuscript fully available?

Reviewer #3: Yes

5. Is the manuscript presented in an intelligible fashion and written in standard English?

Reviewer #3: Yes

6. Review Comments to the Author

Reviewer #3: (No Response)

7. PLOS authors have the option to publish the peer review history of their article (what does this mean? ). If published, this will include your full peer review and any attached files.

**Do you want your identity to be public for this peer review?** For information about this choice, including consent withdrawal, please see our Privacy Policy .

Reviewer #3: No

---

## [Editor Report · Acceptance letter]

PONE-D-23-29484R2

PLOS ONE

Dear Dr. Martín-Fernández,

I'm pleased to inform you that your manuscript has been deemed suitable for publication in PLOS ONE. Congratulations! Your manuscript is now being handed over to our production team.

Kind regards,

on behalf of

Professor Dr Hoh Boon-Peng

Academic Editor

PLOS ONE